# Effect of Doping Microcapsules on Typical Electrical Performances of Self-Healing Polyethylene Insulating Composite

**Youyuan Wang, Yudong Li \*, Zhanxi Zhang and Yanfang Zhang**

The State Key Laboratory of Power Transmission Equipment & System Security and New Technology, Chongqing University, Chongqing 400044, China

\* Correspondence: 20171102031t@cqu.edu.cn; Tel.: +86-1330-837-4882

**Abstract:** Polyethylene cables, as important transmission equipment of modern power grid, would inevitably be slightly damaged, which seriously threatens the safety of the power supply. This paper has pioneered the preparation and typical performances of a self-healing polyethylene insulating composite. The self-healing performance to structural damage was verified by tests of electrical and mechanical damage. The effect mechanism of doping microcapsules on the electrical performance of polyethylene was emphatically analyzed. The results show that in appropriate conditions (such as 60 °C/30 min), the composite can not only repair the electrical tree and scratches, but also restore the insulation strength of damaged area. The effect of doping microcapsules on the electrical performances of polyethylene, such as breakdown strength, volumetric resistivity, dielectric properties, and space charge characteristics, are mainly related to impurity and the interface of microcapsule. Polarization and ionization of impurities can reduce the electrical performance of polyethylene. The interface not only improves the microstructure of polyethylene (such as how the heterogeneous nucleation effect increases the number of crystal regions, and the anchoring effect enhances the stability of amorphous regions), but also increases the charge traps. Moreover, the microstructure and charge trap can affect the characteristics of carrier transport, material polarization, and space charge accumulation, thus improving the electrical performance of polyethylene. In addition, the important electrical performance of the composite can meet the basic application requirements of polyethylene insulating material, which has good application prospects.

**Keywords:** self-healing; polyethylene; microcapsule; crystallinity; electrical tree; breakdown strength; volume resistivity; dielectric properties; space charge

## 1. Introduction

Polymer composites are increasingly used as the main components in various engineering applications. However, micro structural defects are the major damage form of polymer composites, and the damage is usually formed deep within the material, where detection and repair are difficult [1–3]. Moreover, the micro structural defects lead to the degradation of material performances, shortening the service lifespan of material. In the field of electrical insulating materials, reliable and durable insulating polymers are required for high voltage transmission in the power grid [4]. Cross-linked polyethylene (XLPE) insulated cables have been widely used in modern power grids due to their excellent electrical performances. The XLPE cable has become the main transmission equipment of urban power grid [5–7]. However, in the process of manufacture, laying, and operation, damage can inevitably occur in the polyethylene insulation layer, resulting in structural defects, such as micro-cracks and micro-holes [7–9]. Moreover, the outer surface of polyethylene at the joint is also easily scratched [9].

In the high electric field, micro structural damage can distort the electric field, which leads to partial discharge (PD) and electrical trees, causing new damage. Furthermore, the structural degradation can lead to the irreparable decline of material typical performances, such as breakdown strength [10,11]. However, with the existing technologies it is difficult to detect and repair these micro damage defects in polyethylene [12–15]. Therefore, the micro damage can directly threaten the normal operation of the power grid, and even cause catastrophic failures. If the insulating material has self-healing ability, that is, the material can repair the defect automatically at the initial stages of damage, the influence of structural damage on the insulating material will be greatly reduced. However, the existing research on self-healing materials rarely involve the high-voltage insulating material and its electrical performances [16–20]. Therefore, it is necessary to develop a self-healing polyethylene insulating composite and analyze its typical electrical performances, such as insulation strength, dielectric properties, and space charge characteristics. On the premise of guaranteeing the basic performances of polyethylene, the ability to self-heal damage is realized, fundamentally prolonging the service lifespan of insulated cables.

Nowadays, self-healing technologies are mainly divided into two types: intrinsic and extrinsic [18–22]. In recent years, the research on self-healing electrical materials have focused on the intrinsic method [20,21]. However, the intrinsic self-healing materials with grafting groups or the Diels–Alder reaction not only introduce many polar groups, but also require stricter reaction conditions [23–25]. Furthermore, the new self-healing soft electronics, such as dielectric elastomers, usually have a higher dielectric constant and smaller Young's modulus [26,27]. Therefore, the intrinsic self-healing electrical materials cannot meet the application requirements of polyethylene high-voltage insulating materials. Compared with the intrinsic material, the extrinsic self-healing material has a more mature manufacturing process and better weatherability [22]. Thus, the extrinsic method is more conducive to the self-healing behavior of polyethylene insulating materials in complex operating environments. The extrinsic self-healing technologies mainly include microcapsules, hollow fibers, microvascular network, and mesoporous hollow microsphere [28–30]. Moreover, the microcapsule system not only has good stability and a high repair rate, but also has little damage to the structure of the matrix material. Therefore, the microcapsule system was selected to achieve the self-healing goal for the damage in insulating materials. The microcapsule technology refers to using the repairing agent encapsulated in microcapsules to repair the defect in the matrix [1]. The results of many studies show that at the appropriate size and concentration, the microcapsule can withstand the effects of high temperature and stress, and has better repair ability [1,22,31]. Moreover, the microcapsule in composite can concentrate stress, thus attracting and repairing the crack defect [17,19]. However, these studies pay more attention to the mechanical performances of materials, and do not involve the field of electrical insulating materials. In addition, the analysis of the typical physicochemical and electrical performances of materials before and after doping microcapsules are lacking.

At present, there are few reports about external self-healing insulating materials [18–20], and the application prospects in the electricity field are not considered in most of the research on microcapsules [20,22]. In the aspect of simulation, the static stress distribution of microcapsules used for epoxy resin insulating material was simulated [32]. The results confirmed that the internal stress of insulating material can lead to the rupture of microcapsule, thereby completing the repairing behavior. However, it does not involve the repairing ability of actual insulating material. In terms of experiments, the ability of microcapsules to repair electrical tree damage in epoxy resin was studied [33]. The results showed that the microcapsules can not only repair the damaged area, but also delay the development of electrical trees, but the change rules and mechanism of typical performances are still insufficient. In addition, the migration behavior of superparamagnetic nanoparticles was used to repair the electrical tree damage in polymers, while ensuring that the basic electrical properties of the material were not affected [34]. However, this method requires the stimulation of the external magnetic field, which belongs to the inductive repair system, and there is still a certain gap with the active self-healing target of insulating materials. Therefore, the research on the self-healing polyethylene insulating material remains to be deepened.

In this paper, the preparation methods of microcapsule and polyethylene composites were creatively explored and improved to meet the self-healing ability requirements of insulating material. Based on the successful preparation of microcapsule/polyethylene insulating composites, the ability to self-heal structural damage, such as mechanical and electrical damage, was proven by electrical tree, scratch, and breakdown tests. Moreover, the appropriate self-healing conditions of composites were explored. Furthermore, considering the application requirements and performances of polyethylene insulating materials, the crystallinity, volume resistivity, permittivity, and space charge characteristics were selected as the important typical performances to study. The change mechanism of typical performances was innovatively studied from the preparation technology of polyethylene and the doping of microcapsules. In addition, the effect mechanism of doping microcapsules on the typical electrical performances of polyethylene was emphatically analyzed, based on the crystallinity of the matrix, interface, and impurity characteristics of the microcapsule. The pioneering work in this paper can provide an experimental and theoretical basis for optimizing the performances of self-healing insulating material. Besides, the results can be widely applied to plastic polymer insulating materials. Furthermore, this study can provide new ideas for the development of solid insulating materials and self-healing materials.

## 2. Materials and Methods

### 2.1. Materials and Preparation

2.1.1. Preparation of the Microcapsule for Insulating Material

Dicyclopentadiene (DCPD) was selected as the repairing agent (core material) to meet the needs of repairing damage in the polyethylene insulating material. On the one hand, DCPD has the faster reaction rate, which can repair the damage defect in time, on the other hand, the reaction product of DCPD is polydicyclopentadiene (PDCPD). Its dielectric constant and density are close to those of polyethylene, which can effectively homogenize the local high field of the defect, thus reducing the impact of structural damage on polyethylene [14,22]. Therefore, the microcapsule system of urea-formaldehyde (UF) resin encapsulated DCPD was prepared.

In this paper, the UF/DCPD microcapsule were prepared by "two-step" in-situ polymerization to improve the controllability of the reaction, and obtain a microcapsule with appropriate morphology and properties, so that the microcapsule could adapt to the electrical insulation environment. The raw materials used were all analytical reagent (AR) level.

In step one, the urea was dissolved in deionized water, and the formaldehyde solution was added. Then, the pH of solution was adjusted to 8.0~9.0 by triethanola mine (TEA). The UF prepolymer was obtained at 70 °C for 1 h.

In step two, the stable DCPD oil-in-water (O/W) emulsion was obtained by mixing a sodium dodecyl benzene sulfonate (SDBS) emulsifier, melted DCPD, and deionized water for 30 min. Subsequently, the UF prepolymer, the UF curing agent ammonium chloride, and the UF water resistant modifier resorcinol were added to the emulsion. After that, the pH of the emulsion was slowly adjusted to about 3.0 by dilute hydrochloric acid. Finally, the microcapsules were obtained by reaction at 60 °C for 3 h.

In order to make the performances of microcapsule meet the requirements of the insulating material, some variables needed to be regulated. The basic performances of a microcapsule, such as surface morphology and size, were modified by regulating the wall and morphology of microcapsule.

In the aspect of the microcapsule wall, above all, the mole ratio of urea to formaldehyde was 1. This ensured a sufficient content of dimethylol urea in the UF prepolymer, which could enhance the net structure of UF wall. Moreover, the high mole ratio could increase the toughness of the UF wall, so that the microcapsule could resist the normal thermal, electrical, and mechanical stresses in polyethylene insulating material. Then, the dosage of ammonium chloride and resorcinol were 1.5% and 2.5% of the prepolymer, respectively, to enhance the stability of the microcapsule wall structure.

In addition, the acidification time (i.e., reaction rate of the wall material) was controlled at 20~30 min to obtain a better spherical shell structure.

In the aspect of microcapsule morphology, the speed of emulsification was 400 rpm, to adjust the size of the core droplets (about 100 μm). Thus, the size of microcapsule suitable for polyethylene could be obtained, which ensured the repair effect and reduced the influence on the performance of the insulating material [31,32]. Moreover, the dosage of SDBS was 5% of the DCPD mass, to ensure the dispersion and stability of emulsion. Besides this, the mass ratio of UF prepolymer to DCPD was about 2:1, to ensure the appropriate encapsulation effect. Furthermore, the rotational speed was adjusted to 300 rpm after acidification to obtain a microcapsule with intact sphericity and excellent dispersibility.

### 2.1.2. Preparation of the Novel Self-Healing Polyethylene Insulating Composite

At present, there is no research on self-healing microcapsule/polyethylene insulating composites. Thus, it is novel research to use a microcapsule system to automatically repair the structural damage of polyethylene insulating material.

Low-density polyethylene (LDPE), as the main raw material of XLPE, has few impurities. Moreover, LDPE is also the main raw material of various functional polyethylenes. Therefore, the microcapsule/LDPE insulating composite was taken as the research object. The raw material of polyethylene is 2426H LDPE. The LDPE particles were ground into LDPE powder (15 meshes), to ensure the uniform mixing of the polyethylene and microcapsule system.

According to relevant literature [17] and my previous work [35] (as shown in Table 1), the basic properties (such as thermal stability and tensile strength) of a composite doped with about 1 wt.% microcapsule are more in line with the application requirements of the cable. It has the best thermal stability and better tensile strength, on the premise of a higher repair efficiency. Preliminary analysis showed that the effect of microcapsules on the basic properties of polyethylene is mainly related to the interface and impurities introduced by the microcapsule. Moreover, an appropriate number of interfaces can improve the properties of the material (such as thermal stability), but excessive impurities and interface defects can reduce its properties (such as tensile strength and thermal stability) [35]. Therefore, the performance of the composite is affected by the concentration of microcapsule. Overall, when the concentration of the microcapsule is about 1 wt.%, the comprehensive properties of the composite are better. Thus, the more appropriate concentration (1 wt.%) was selected to study the effect mechanisms of microcapsules on the electrical performance of polyethylene and exclude the influence of other secondary factors. In addition, Grubbs' second-generation catalyst, with better stability and catalytic efficiency, was selected to improve the effect of self-healing, and its concentration accounted for 10 wt.% of the microcapsule.

**Table 1.** Basic performance of the composite with different concentrations of microcapsule.

| Basic Performances of Composite | Concentrations of Microcapsule in Composite | | | | |
| --- | --- | --- | --- | --- | --- |
| | 0 wt.% | 0.5 wt.% | 1 wt.% | 5 wt.% | 10 wt.% |
| Repair efficiency for scratches about 100 μm wide | 0% | ≥56.3% | ≥82.4% | ≥85.1% | ≥90.0% |
| Thermal decomposition temperature | 442.9 °C | 445.1 °C | 456.7 °C | 438.0 °C | 432.0 °C |
| Tensile strength | 11.62 MPa | 11.58 MPa | 11.14 MPa | 9.11 MPa | 3.61 MPa |

The microcapsule/LDPE composite was prepared by the melt blending method [36]. Due to the demand of self-healing behavior, the microcapsule needed to have the characteristic of rupturing under the stress condition. In the original preparation condition, the microcapsule could be ruptured (as shown in Figure 1a, thus, the self-healing ability could not be realized. However, if the thickness and strength of the microcapsule wall structure are excessively increased, the microcapsule may not respond to the damage of matrix in time. Therefore, the thermal stress and mechanical stress in the original preparation process have been adjusted in this paper.

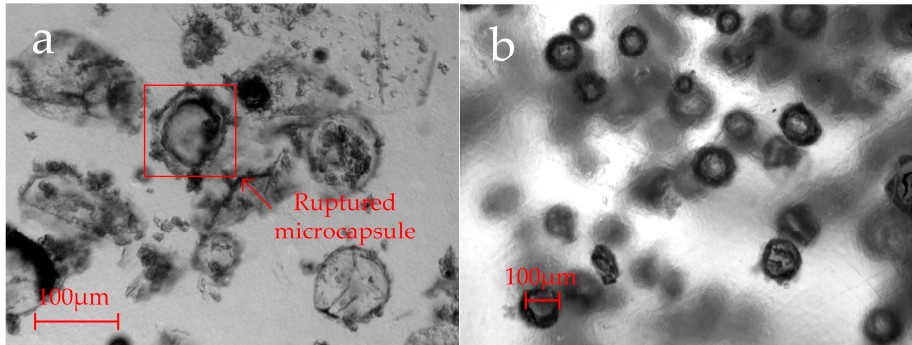

**Figure 1.** Microstructure of the microcapsule/LDPE (low-density polyethylene) insulating composite samples. (**a**) Original technology. (**b**) Modified technology.

The composite was obtained by the Hangfa YJ8F flat vulcanizing machine. The working pressure was 3 MPa, which was designed to avoid damage to the microcapsule, while ensuring sample formation. The working temperature was 160 °C, and the pressing time was 30 minutes, to further enhance the morphology of sample. In Figure 1b, the morphology of samples with modified technology is better, the microcapsules are uniformly dispersed and have no obvious rupture, which meets the application requirements.

In addition, the pure LDPE sample was prepared according to the original method, to analyze the influence of preparation technology on the typical performance of LDPE. The working pressure was 15 MPa, and the pressing time was 20 minutes [36].

*2.2. Methods*

2.2.1. Verification of Self-Healing Performance

In this paper, the self-healing performance of the composite against structural damage was verified from two aspects: mechanical damage and electrical damage. Furthermore, the effect of the microcapsules on the tree discharge and insulation strength of polyethylene were explored.

In terms of mechanical damage, scratch damage was used to simulate the mechanical structural damage in polyethylene insulating material. Several groups of scratches were carried out with the Tianchuang QHZ scratch tester. Then, the samples were placed in a heating oven to ensure the repair effect. The morphological characteristics of the scratches was observed by optical microscope (OM). Moreover, the alternating current (AC) breakdown strength of samples was carried out to verify the self-healing performance for mechanical damage in electrical insulation more clearly. In addition, the samples were heated at different temperatures and times, to explore the appropriate conditions for the repair reaction. According to the optimum polymerization temperature (about 45 °C) of the repair agent (DCPD), curing temperature (above 60 °C) of the reaction product (PDCPD), and normal operating temperature (less than 90 °C) of the cable, temperatures of 45 °C, 60 °C, and 90 °C were selected in this paper. The heating time was 10 min, 30 min, and 60 min, respectively.

In terms of electrical damage, the self-healing performance of the composite was verified by electrical tree. The needle-plate electrode system was used to initiate the electrical tree, and the distance between electrodes was 5 mm. The needle electrode was inserted into the sample during the preparation process, while the curvature radius of the needle tip was 5 μm. The plate electrode contacted the sample through conductive adhesive. Moreover, the sample was immersed in transformer oil to prevent surface flashover. In the experiment, an AC voltage of 7 kV/50 Hz was applied for 90 min. After that, the samples were placed in suitable conditions for heating. The morphological characteristics of the electrical tree were observed by OM.

### 2.2.2. Characterization

In this paper, the typical electrical performances of polyethylene were characterized by insulation strength, resistivity, dielectric properties, and space charge characteristics. Moreover, the effect mechanisms of the doping microcapsule on the electrical performances of polyethylene were explored.

Polyethylene has both crystalline and amorphous regions, that is, the crystallization characteristics of polyethylene can directly affect its basic electrical performance [37]. The SETARAM-DSC 141 differential scanning calorimeter (DSC) was used to measure the crystallinity of pure LDPE and its composite. The temperature range was 50–160 °C, the heating rate was 10 °C/ min, and the nitrogen flow rate was 150 ml/ min. The mass of the sample taken for this test was 10 mg.

Usually, the higher the resistivity, the higher the efficiency of the material used in electrical insulating components [37]. The volume resistivity of pure LDPE and its composite was tested by the BEST-212 volume resistivity measurement system. The volume resistivity measurement was carried out at room temperature (25 ± 1 °C).

Insulating material needs to have a smaller dielectric constant and lower dielectric loss to ensure normal insulation effects [36]. The dielectric constant (i.e., permittivity) of the samples was measured by the NOVOCONTROL CONCEPT 80 broadband dielectric measurement system. The frequency range was $10^{-1}$–$10^6$ Hz. The measurement was carried out at room temperature.

Space charge is liable to accumulate in polyethylene, which can distort the local electric field and affect its service [38]. The space charge measurement was carried out with the Chongqing University pulsed electro-acoustic (PEA) system. The accumulation characteristic of space charge was measured under a direct-current (DC) voltage of 20 kV/mm, and the acquisition time points were 10 s, 1 min, 5 min, 10 min, and 30 min. This measurement was carried out at room temperature.

In addition, all performance tests were performed at least five times to reduce the contingency and dispersion of the experimental results. In the electrical performance tests, the silicone oil was smeared between the electrode and sample to eliminate the influence of air gap on the signal, ensuring the accuracy of the results.

## 3. Results and Discussion

### 3.1. Self-Healing Performance

### 3.1.1. Repair Effect of Mechanical Damage and Breakdown Strength

Breakdown strength is one of the most important macroscopic performances of insulating materials, which can be affected by damage defects. The scratch defect can be equivalent to mechanical structural damage in polyethylene insulating materials. Therefore, the self-healing ability was visually represented by the AC breakdown strength of the sample before and after repairing damage. The variation in breakdown strength is shown in Figure 2.

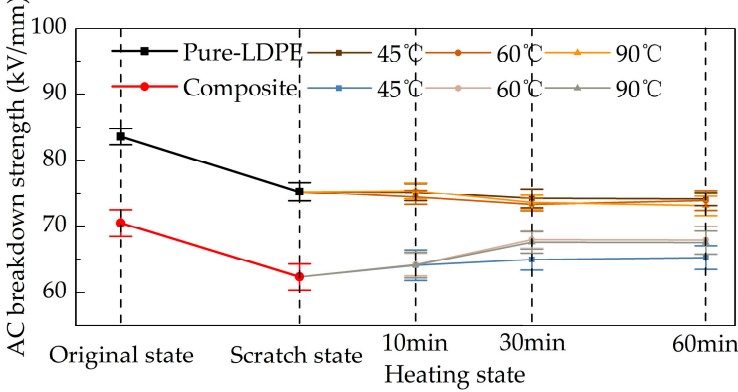

**Figure 2.** Alternating current (AC) breakdown strength of polyethylene samples.

Although the breakdown strength of pure polyethylene decreases slightly after modifying the process, the change trend of its breakdown strength after damage is the same as that of the original polyethylene. In addition, this paper focuses on the effect of microcapsules on the typical electrical performances of polyethylene. Therefore, this paper only gives the mechanical damage and breakdown strength results of pure polyethylene with a modified process, which was the comparison with the composite.

In Figure 2, the insulation strength of polyethylene can be reduced by scratch damage. For pure polyethylene, different heating temperatures and times can only make the breakdown strength fluctuate slightly. In other words, the heating conditions in this paper had almost no effect on the insulation performance of pure polyethylene. For the composite, when heated for 10 min at different temperatures, the breakdown strength was clearly recovered, and the recovery range was basically the same. However, when heated for 30 min, the breakdown strength of the samples at 60 °C and 90 °C can be greatly increased, while that of the sample at 45 °C was smaller. When heated for 60 min, the breakdown strength of all samples tended to change smoothly.

It can be concluded that when the temperature was higher than 45 °C, the DCPD could rapidly initiate the ring-opening shift polymerization reaction, thus filling the damage defects and restoring the material performance, to a certain extent. However, when the temperature was higher than 60 °C, the polymerization product could be cured better, and the properties of the repair product could be improved, so as to further repair the performance of the matrix. The curing time was 10–30 minutes. Therefore, the suitable repair condition selected in this paper was 60 °C/30 min, which fully meets the operating conditions of polyethylene insulating material. In these conditions, the insulation performance of the polyethylene matrix hardly changed, but the self-healing effect was obvious.

This paper focuses on the study of self-healing behavior in the appropriate repair conditions (such as 60 °C/30 min). Figure 3 shows the OM results of the scratch test.

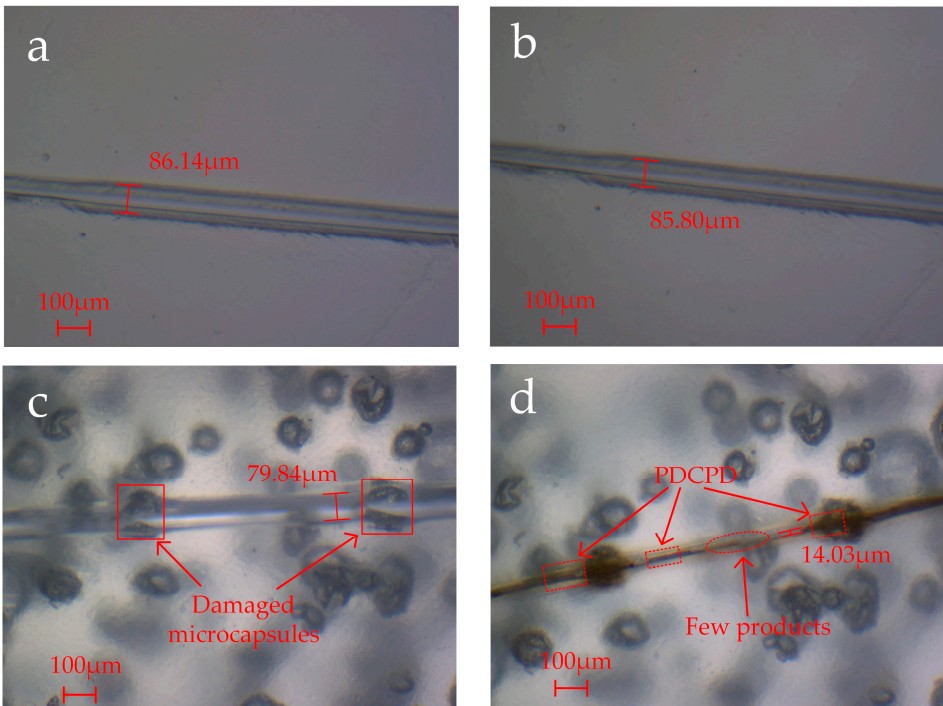

**Figure 3.** Morphology of the scratch. (**a**) Unheated pure LDPE. (**b**) Heated pure LDPE. (**c**) Unheated composite. (**d**) Heated composite.

In pure LDPE, the scratch morphology hardly changed after heating. Scratches not only destroy the physical structure of polyethylene, but also lead to a decrease in breakdown strength. This is mainly caused by the distortion effect of a defective structure on electric field [14]. After heating,

the breakdown strength of pure polyethylene showed no obvious recovery. Thus, structural damage is irreversible in pure polyethylene, which can directly reduce the insulation strength of polyethylene.

Doping microcapsules can reduce the insulation strength of polyethylene to a certain extent, which is mainly related to the introduction of impurities and interface defects. In the process of doping microcapsules, many impurities, such as polar groups, can be introduced, which can be ionized in the electrical field, increasing carrier concentration. In addition, the interface area between the microcapsule and matrix can introduce new structural defects, which can not only increase the number of PD points, but also form relatively concentrated low-density areas [39]. Therefore, doping microcapsules can reduce the breakdown strength of polyethylene and increase its dispersion.

In composites, the scratch can break the wall structure of the microcapsule in the polyethylene matrix and reduce the breakdown strength of the composite, which is similar to that of pure polyethylene. The thermal conditions and capillary action can cause the DCPD core material to flow into the defect site. The liquid DPCD contacts with the catalyst and reacts in the damaged area to form the PDCPD solid, whose thermal deformation temperature is 120 °C [40]. Moreover, the density and dielectric constant of PDCPD are similar to those of polyethylene. Thus, PDCPD can homogenize the local high field of the defect and decrease the local low-density area, reducing the electromechanical stress and the disorder of structure [10,37,39], thus, improving the electrical insulation performance of the matrix. Therefore, compared with pure LDPE, the composite has obvious self-healing effects on the microstructure and breakdown strength of mechanical damage. The repair efficiency for scratch width was 82–100%, and the breakdown strength could be restored to about 94.4% of the original state.

In Figure 3d, the self-healing effect of the damaged microcapsules surrounding area is more obvious, and the crack is almost completely repaired. The main reason is that the damaged area near the microcapsules can preferentially contact a large amount of repairing agent, and the reaction rate between DCPD and the catalyst is relatively fast, causing the DCPD to already begin to react in the diffusion process. The fast reaction speed can better meet the timely requirement of the insulating material for damage treatment.

At present, the research on self-healing insulating material is still in the exploratory stage. The experimental results show that the composite prepared in this paper does have the ability to self-heal mechanical structural damage. Although a small part of insulation strength is sacrificed, the composite can quickly restore the insulation performance after damage. Thus, the composite still has good research value in lower voltage application scenarios.

### 3.1.2. Repair Effect of Electrical Damage

For high voltage and ultra-high voltage polyethylene cables, electrical tree is the main factor leading to insulation failure [41]. In the electric field, contaminants (impurities) and defective structures (such as protrusions and voids) in polyethylene cables can distort the electric field, resulting in electrical damage, such as PD and electrical tree. Electric tree is a degradation process, which can cause irreversible structural damage of microns and above, and directly threaten the insulation strength of polyethylene [41,42]. Therefore, the self-healing of electrical tree can greatly improve the insulation performance and service life of polyethylene, but there are still few reports about it. This paper mainly studies the propagation stage of electrical tree. The experimental results are shown in Figure 4. The heating condition is 60 °C/30 min.

This paper observes the panorama of electrical tree by a mosaic of multiple graphs to ensure clarity. In addition, there was no significant difference in the electrical tree of pure polyethylene prepared by the two processes. Furthermore, the emphasis of this paper is to study the effect of doping microcapsules on the typical performance of polyethylene. Therefore, this paper only gives the results of electrical tree in pure LDPE, with a modified process, as a comparison with the composite.

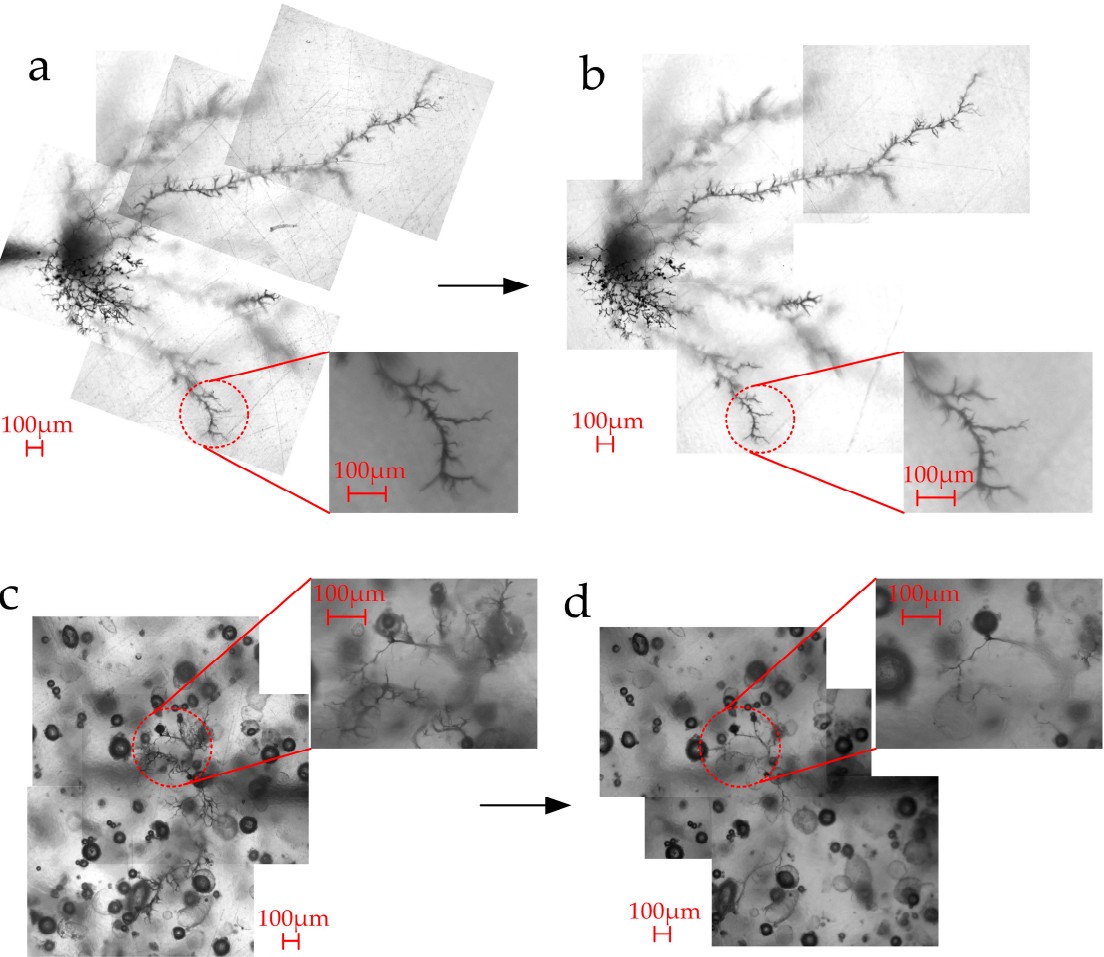

**Figure 4.** Morphology of electrical tree. (**a**) Unheated pure LDPE. (**b**) Heated pure LDPE. (**c**) Unheated composite. (**d**) Heated composite.

Compared with pure polyethylene, there were three distinct changes in the morphology of electrical tree in the composite: 1. the size of electrical tree decreased significantly, which is similar to the result in [33]; 2. the electrical tree can develop into microcapsules and stop growing; 3. the electrical tree tended to branch near the microcapsule. It can be inferred that the local electric field can be affected by the microcapsule and the interface area between the microcapsule and matrix, which can attract the propagation of the electrical tree. When the electric tree develops into the microcapsule, it can break the wall structure of the microcapsule and consume some energy. The PD generated by the residual energy will turn into a new branch, thus the propagation of the electric tree is inhibited to a certain extent. Moreover, the microcapsule and interface area can destroy the continuity of the original structure in polyethylene, thus hindering the propagation of electrical tree. Therefore, the overall size of the electrical tree in the composite decreased and tended to develop towards the microcapsule area.

In the appropriate conditions (60 °C/30 min), the composite exhibited an obvious self-healing ability for electrical damage defects. The morphology of electrical tree in the pure polyethylene had no obvious change. In the composite, except for the main branch of electric tree, the other branches disappeared. Moreover, the width of the main branch was clearly decreased. The electrical tree attracted by the microcapsule can break the wall of the microcapsule, while the core material of the microcapsule can flow into the tubules of electrical tree through capillary and infiltration action. The repair agent contacts and reacts with the catalyst to fill the damage defect. In addition, due to the density and dielectric constant of the reaction product being close to those of polyethylene, the compatibility between PDCPD and matrix is good. Thus, the low-density areas can be reduced.

Due to the large size of the electrical tree, the repair agent cannot completely repair the tubules of the electrical tree. However, the temperature and electric field coexist in the actual operation of polyethylene insulating material, and the propagation time of the actual electrical tree is longer [33,41], which provides a good opportunity for the repair behavior of the microcapsule. In other words, the repair agent in the microcapsule can begin to repair the electrical tree at the early stages of its propagation. Therefore, when the doping amount of the microcapsule is appropriate, the polyethylene insulating composite is fully capable of repairing electrical damage, such as electrical trees.

For a long time, electrical trees have been regarded as irreversible, permanent damage [34]. In this paper, the electrical tree formed by the faults/contaminants in polyethylene could be prevented/delayed by the microcapsule. The microcapsule could not only attract the electrical tree, and inhibit its propagation process, but also repair the tubules of the electrical tree effectively. It has good application value in reducing the harm of electrical damage.

### 3.2. Analysis of Typical Performances

### 3.2.1. Crystallization Characteristics

The crystallinity of polyethylene can directly affect its various properties, but the effect of doping microcapsules on the crystallization characteristics of polyethylene is not clear. Therefore, the crystallization characteristics of samples were analyzed in this paper. Figure 5 shows the heat flow curve of the samples. The crystallinity can be calculated by Formula (1) [43]:

$$X_c = \frac{\Delta H_m}{(1 - \omega)H_N} \times 100\%, \tag{1}$$

where $X_c$ is the crystallinity of the polymer, $\Delta H_m$ is the melting heat, $H_N$ is the melting heat when the crystallinity of polyethylene reaches 100%, and it is 293.6 J/g, $\omega$ is the mass fraction of microcapsule in composite.

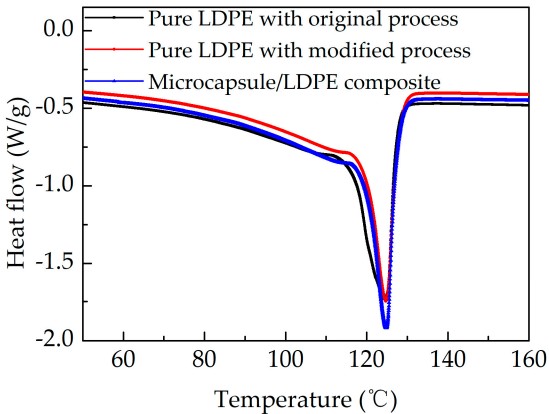

**Figure 5.** Heat flow curves of samples.

The results are shown in Table 2, where $T_p$ is the peak melting temperature and $\Delta T_p$ is the half-height width of the melting peak. The crystallinity ($X_c$) of polyethylene decreased with the change in preparation technology. However, the microcapsule could increase the crystallinity, and the crystallinity of the composite was the highest. The melting peak ($T_p$) was negatively correlated with the flexibility of the polymer. Moreover, the polymer with higher crystallinity had a tighter internal structure and smaller free volume, so the flexibility of the molecular chain was lower [36]. Therefore, with the increase in the crystallinity, the $T_p$ was increased. The half-height width of the melting peak ($\Delta T_p$) denotes the distribution of crystal grain size in polyethylene. The distribution of

grain size in polyethylene can be more uniform by changing the preparation process, and it can be more concentrated by the doping microcapsule.

**Table 2.** Crystallinity and melting peak of samples.

| Samples | $T_p$ (°C) | $\Delta T_p$ (°C) | $\Delta H_m$ (J/g) | $X_c$ (%) |
|---|---|---|---|---|
| Pure LDPE with original process | 124.71 | 7.77 | 123.1 | 41.93 |
| Pure LDPE with modified process | 124.57 | 5.75 | 119.6 | 40.74 |
| Microcapsule/LDPE composite | 124.82 | 5.46 | 129.1 | 44.42 |

Compared with different preparation processes, the change in crystallinity was mainly related to the working stress magnitude and melting time during processing. At the action of stress, the orientation occurs along the direction of external force. The orientated structure can induce nucleation, which accelerates the nucleation rate, increases the number of crystal nuclei, and shortens the crystallization time [44]. Therefore, the original process with higher working stress has higher crystallinity. High temperatures can destroy the original crystalline structure of the polymer [36,44]. The amount of destroyed crystalline structure is increased after prolonging the heating time, and the material tends to undergo a "homogeneous nucleation" crystallization process when the sample is cooled. Hence, the crystallization speed becomes slower and the distribution of grain size is more uniform.

The decrease in crystal nucleus number and the destruction of the crystalline region can directly decrease the crystallinity. Moreover, the decrease in crystallization rate can reduce the grain size. In addition, the homogeneous nucleation can make the distribution of grain size more uniform, which leads to the decrease in $\Delta T_p$. Therefore, as shown in Figure 6, changing the preparation process of polyethylene can reduce the number of large-size crystal grains and produce a few smaller crystal regions, thus reducing the crystallinity of polyethylene.

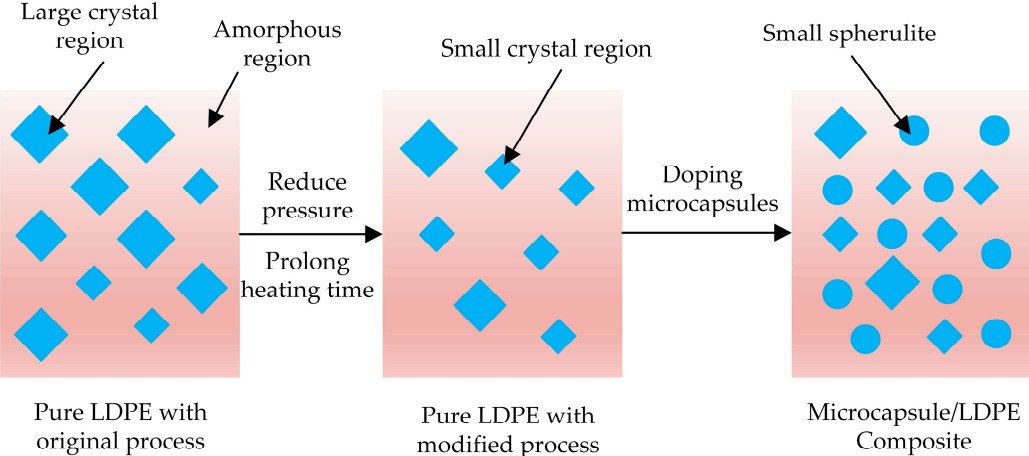

**Figure 6.** The changes in crystallinity in the polyethylene material.

Microcapsules can introduce many interface regions in the polyethylene matrix. The interface between the microcapsule and matrix plays a role of "heterogeneous nucleation" in the recrystallization process. Hence, the interface improves the crystallization rate and reduces the growth rate of spherulites, resulting in many uniform and orderly small spherulites [45,46]. As shown in Figure 6, the microcapsules can increase the number of small size crystal regions in polyethylene and make the distribution of grain size more concentrated. However, microcapsules cannot change the size of the original crystal regions in the matrix. Therefore, the microcapsule increases the crystallization source of polyethylene matrix, resulting in the increase in crystallinity.

In general, doping microcapsules can reduce the crystal grain size, increase the grain number, and concentrate the distribution of grain size, thus affecting the crystallization characteristics of

polyethylene. On the one hand, the reduction in preparation pressure can reduce the efficiency of induction nucleation, and the prolongation of preparation time can destroy the grain structure, leading to the homogeneous nucleation of polyethylene. On the other hand, the interface between the microcapsule and matrix can enhance heterogeneous nucleation, increasing the number of spherulites in the polyethylene matrix.

### 3.2.2. Resistance Characteristic

The volumetric resistivity of the polyethylene composite can reflect the overall insulation performance of the material. Volume resistivity refers to the impedance effect of a material per unit of volume to electric current, as shown in formula (2). The volume resistivity is shown in Figure 7.

$$\rho_v = R_v \frac{S}{h}, \tag{2}$$

where $\rho_v$ is the volume resistivity, $R_v$ is the resistance value, $h$ is the thickness of the sample (i.e., the distance between the electrodes), and $S$ is the area of the electrode.

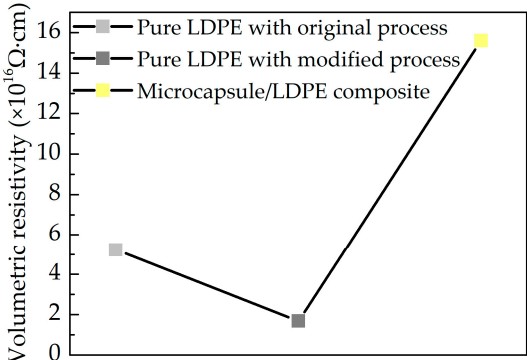

**Figure 7.** Volume resistivity of the samples.

The variation trend of the volume resistivity is consistent with that of the crystallinity (Section 3.2.1). Although the volume resistivity of polyethylene is decreased after changing the preparation technology, the microcapsule can increase its volume resistivity. Moreover, the volume resistivity of the composite is the largest.

Volume resistivity is a characterization of electrical conductivity that is usually independent of material size [47]. The ionic conductivity is the main conductive form of the polymer. Furthermore, space charge traps can be formed in the interface between the crystal and amorphous regions, whose trapping effect on carriers can reduce the conductivity of the material [37,47]. In other words, the larger the crystallinity, the smaller the ionic conductivity. In addition, the microstructure in high crystallinity material is more stable and less affected by electric field and injected charge. Thus, the volume resistance of polyethylene is positively correlated with its crystallinity.

Doping microcapsules can introduce new interface regions, which not only contain many charge traps [37,46], but also increase the number of crystal grains in the matrix (Section 3.2.1)—that is, increase the interface between the crystal and amorphous regions. Therefore, doping microcapsules can increase the charge traps in polyethylene. Charge traps formed in many interface regions can capture carriers (as shown in Figure 8), reducing their movement rate and shortening their free path, thus increasing the volume resistivity.

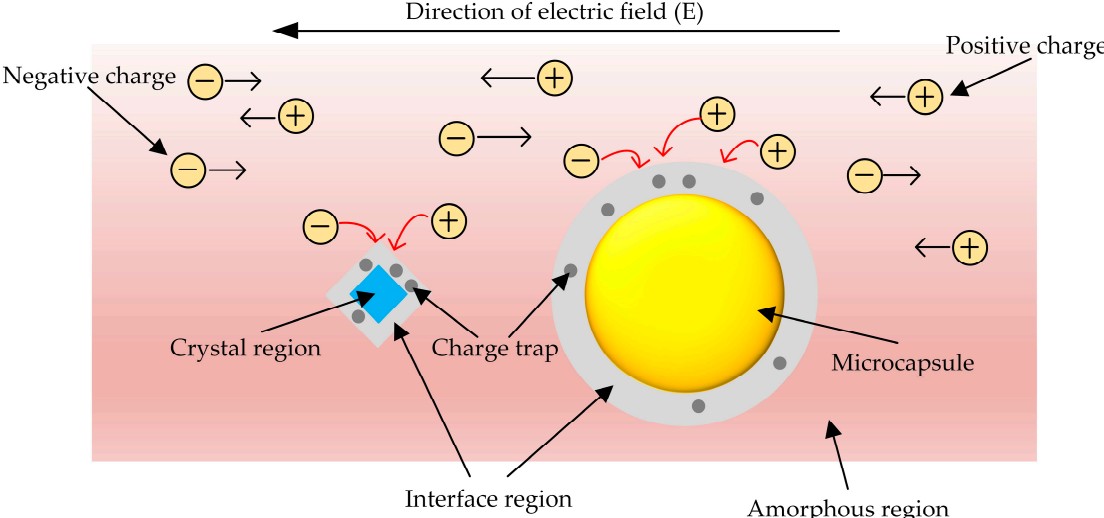

**Figure 8.** Effect of the charge traps in the interface region on carriers.

In brief, the crystallization characteristics of polyethylene can affect its conductivity by changing its ionic conductivity and carrier transport. Moreover, doping microcapsules can not only increase the crystallinity of polyethylene, but also introduce new charge traps. Furthermore, due to the small doping number of microcapsules, the influence of their resistance characteristics on the polyethylene matrix is not obvious. Therefore, doping 1 wt.% microcapsules can greatly increase the volume resistance of polyethylene. Besides this, the higher volume resistance of the composite can meet the application requirements of polyethylene insulating material.

### 3.2.3. Dielectric Properties

Dielectric properties (such as permittivity ε) are an important index for characterizing the electrical performances of dielectric or insulating materials. Permittivity is mainly divided into real and imaginary parts. The real part of permittivity ($\varepsilon'$) characterizes the polarization property of the material in the electric field. Moreover, the imaginary part of permittivity ($\varepsilon''$) characterizes the energy loss (i.e., dielectric loss) in the process of polarization. Figure 9 shows the dielectric properties of samples.

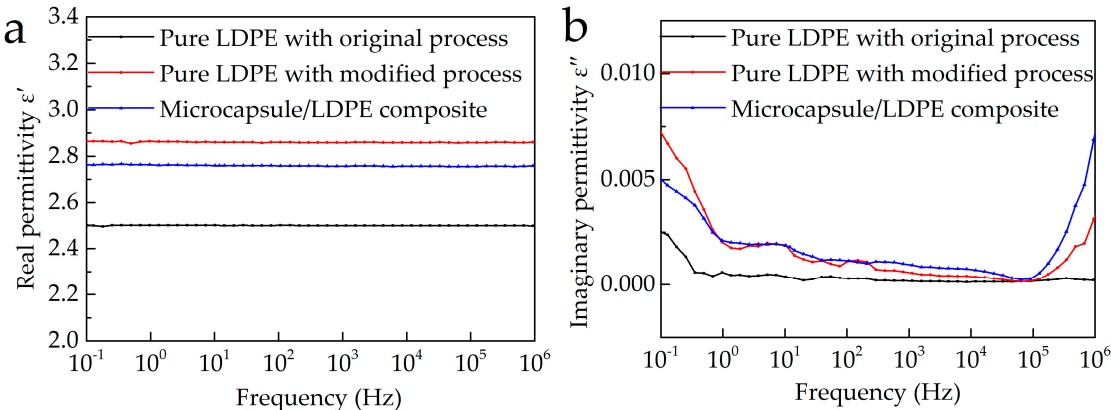

**Figure 9.** Dielectric properties of samples. (**a**) Real part of permittivity. (**b**) Imaginary part of permittivity.

Polyethylene belongs to the non-polar polymers. The polarization of polyethylene is mainly electron polarization and atomic polarization. Due to its weak polarization property, its permittivity is small. The $\varepsilon'$ of LDPE is clearly increased after changing the preparation process. Compared with the pure LDPE with modified process, the $\varepsilon'$ of the composite is decreased slightly, but it is still higher than that of the original sample. In addition, the $\varepsilon'$ of all samples is increased slightly with the decrease in

frequency. This phenomenon is caused by the more sufficient polarization behavior of the dielectric in low-frequency regions, such as electron polarization and impurity polarization [48].

Dielectric loss is mainly caused by the steering polarization of the material. The non-polar polymer is theoretically without steering polarization. In other words, there is no dielectric loss in polyethylene. However, due to the impurities and structural defects in the material, current leakage can be caused [36,39]. Thus, a part of the electrical energy is transformed into the heat energy, which is the conduction loss. Therefore, the dielectric loss of polyethylene in the low-frequency regions is mainly conduction loss. All samples have obvious dielectric loss in the low-frequency regions. Moreover, the dielectric loss of LDPE in low-frequency regions is the largest after changing the process. Compared with the pure LDPE with modified process, the dielectric loss of the composite is decreased slightly.

In the intermediate-frequency regions, the $\varepsilon''$ of LDPE is increased slightly by changing the preparation process or doping microcapsule. In addition, the dielectric loss is decreased with the increase in frequency.

In the high-frequency regions, the $\varepsilon''$ of LDPE is increased by changing the preparation process. Moreover, the dielectric loss can be increased further after the doping microcapsule.

The DC conduction loss can be neglected in high-frequency regions [49]. Moreover, the steering polarization of polar molecules cannot be established in high-frequency regions. At the electric field, the charge traps in the material can capture, emit, and re-capture the carriers, and release energy in the form of photons and phonons, thus causing the dielectric loss [37,49]. The higher the frequency, the faster the process of capturing and releasing the carriers by the charge traps, so the more energy is released. This leads to the increase in dielectric loss with the increase in frequency. Therefore, the dielectric loss of polyethylene in the high-frequency band is mainly the relaxation polarization of the charge traps.

Compared with the different preparation processes, the crystallinity of LDPE with a modified process is decreased, loosening the internal structure of the sample. Loose internal structure can reduce the hindrance to the movement of molecular chains and the steering polarization of the polar impurity group. Thus, the polarization property of polyethylene is enhanced, increasing the $\varepsilon'$ (as shown in Figure 9a). Moreover, the steering polarization of molecular chains and impurities can cause energy loss, resulting in an increase in the $\varepsilon''$ in middle and low frequencies (as shown in Figure 9b). However, with the increase in frequency, the steering polarization cannot be established, and its energy loss becomes smaller. Therefore, the $\varepsilon''$ of LDPE is decreased with the increase in frequency.

In addition, with the decrease in crystallinity, the free volume of material is increased. Hence, the micro-structure becomes rough and loose, increasing the internal defects, which increases the depth and density of charge traps in the sample. Therefore, the relaxation loss of deep traps in LDPE with modified process is increased, and the $\varepsilon''$ rises in high-frequency regions.

Microcapsules can improve the crystallinity of the LDPE matrix, making the internal structure more compact and orderly. Moreover, the interface region between the microcapsule and LDPE matrix anchors the macromolecular chains in the amorphous region of the material (as shown in Figure 10). The anchoring effect can hinder the movement of the molecular chains or impurity groups, improving the stability of the internal structure. Thus, the steering polarization of molecular chains and polar impurity groups in the amorphous region of composite is weakened. In other words, the increase in crystallinity and the anchoring effect of the interface can reduce the polarization property and conduction loss of polyethylene matrix. Therefore, the $\varepsilon'$ in all frequency bands and the $\varepsilon''$ in the low-frequency region of the composite are decreased.

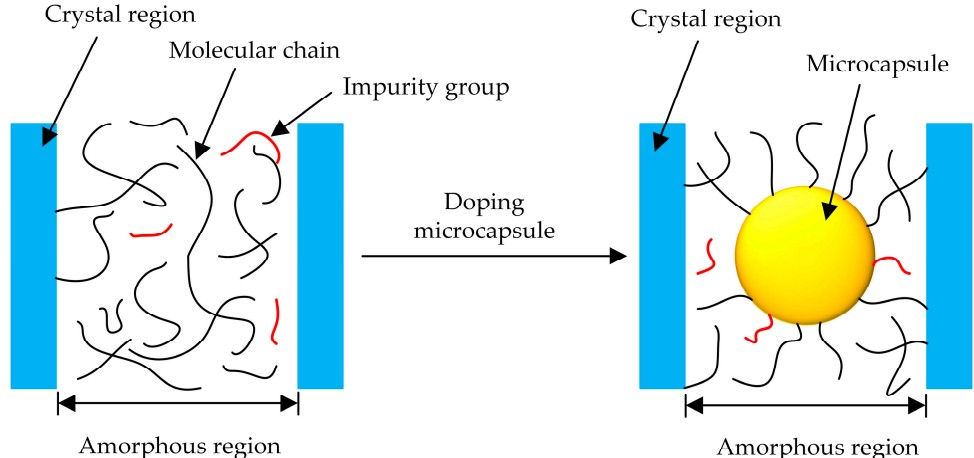

**Figure 10.** Anchoring effect of microcapsule in polyethylene.

The interface polarization (i.e., space charge polarization) induced by the interface region between the microcapsule and the matrix was enhanced [46,50]. Moreover, the polarization effect of the microcapsules and the catalyst particles themselves cannot be ignored. Therefore, the $\varepsilon'$ of the composite cannot be restored to the original state, and its dielectric loss in the low- and medium-frequency regions is also higher.

In addition, the interface region introduced by the microcapsule system can lead to many localized states near the conduction band and the valence band. As a result, the interface region can introduce many new charge traps (as shown in Figure 8), which leads to the increase in traps relaxation loss in the material. Therefore, the $\varepsilon''$ of the composite is increased further in the high-frequency regions.

Overall, doping microcapsules can change the structural characteristics of crystalline and amorphous regions in polyethylene, and introduce impurities and interfaces. These can affect the polarization process of polyethylene, thus changing the dielectric constant and dielectric loss of the polyethylene matrix. The $\varepsilon'$ of the composite does not fluctuate significantly in all frequency bands, and its dielectric loss (i.e., $\varepsilon''$) is maintained at a relatively low level. Therefore, the composite can meet the basic application requirements of insulating material.

### 3.2.4. Space Charge Characteristics

In DC transmission, space charge can accumulate in polyethylene. Excessive local charges can distort the local electric field and accelerate the insulation deterioration, thus reducing the reliability of the cable [51,52]. Because the distribution of space charge accumulation is directly related to the weak parts of the insulating material, this paper focuses on the space charge accumulation characteristic. The test results are shown in Figure 11. The mean volume density of space charge by formula (3) was calculated to compare the accumulation of space charge more intuitively [36]:

$$q\big(t; E_p\big) = \frac{1}{x_1 - x_0} \int_{x_0}^{x_1} \big|q_P(x, t; E_P)\big| dx, \tag{3}$$

where $q(t;E_p)$ is the mean volume density of space charge, $x_1$ and $x_0$ are the positions of the anode and cathode, respectively, $t$ is the polarization time, $E_p$ is the polarization field strength, and $q_p(x,t;E_p)$ represents the space charge density in the material.

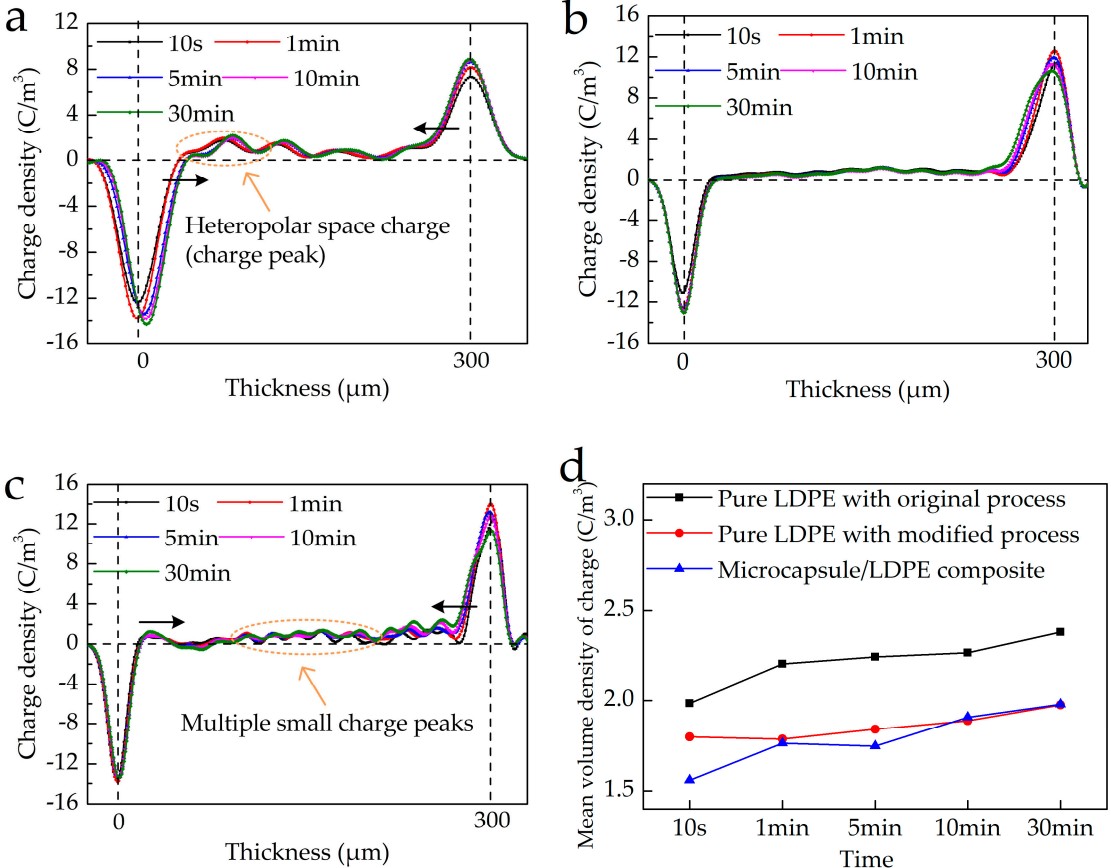

**Figure 11.** Space charge accumulation characteristics of polyethylene. (**a**) Pure polyethylene with the original process. (**b**) Pure polyethylene with the modified process. (**c**) Microcapsule/polyethylene composite. (**d**) Mean volume density of space charge.

The accumulation of a positive space charge was observed in all samples. This may be mainly related to the DC bias pulse power supply and the errors of the test instruments. However, the trends of space charge accumulation are the same for many groups of test results, which does not affect the results analysis.

In pure polyethylene with the original process, there was an obvious accumulation of heteropolar space charge near the electrode. With an increase in test time, the space charge peak (space charge packet) increased gradually, and migration to the interior of the sample and the mean volume density of the space charge increased gradually.

The main sources of space charge in polyethylene are electrode injection and impurity ionization [50,51]. Although polyethylene itself is a non-polar molecular structure, the polar impurities are introduced inevitably in the preparation process. In high electric fields, polar groups are ionized and decomposed, then attracted by electrodes and accumulated near the electrodes to form heteropolar charges. Moreover, due to the large mass of ionized products and the limitation of localized state in the dielectric, the migration rate of ions is slow, which makes it easy for ions to form charge packets. On the other hand, the electrodes can inject carriers, which move to the vicinity of their heteropolar electrodes to form heteropolar charge. In the movement process of the carrier, the repulsion force of the same polar charge packet can slow down the carrier motion. Then, the carrier is trapped by the charge trap near the charge packet, thus making the charge peak increase and move into the sample. It can be inferred that the charge trap depth of pure polyethylene with the original process is not large, so the carrier injected by the electrode has a greater probability of moving into the material.

Compared with the original process, the accumulation of heteropolar charge near the electrode in the pure polyethylene with the modified process was significantly reduced. The charge accumulation

(space charge peak) in the sample was not obvious. Moreover, the mean volume density of space charge was kept at a low level.

The mobility of the carrier in the dielectric was not only related to the micro-parameters, such as charge traps, but also to the macro-parameters, such as volume resistivity. Its behavior determined the accumulation characteristics of space charges in the material [37,51]. The volume resistivity of the pure polyethylene with the modified process clearly decreased (Section 3.2.2), so the barrier effect on carriers was reduced, that is, the it was easier for the charge to reach the electrode. Moreover, because the grain size and amount of the sample were decreased (Section 3.2.1), the interface between the crystal and amorphous regions was decreased. Thus, the charge trap formed by localized state in the interface was decreased [37,46], which reduced the binding capacity of polyethylene to space charge. Therefore, in the pure polyethylene with modified process, the charge from electrode injection and impurity ionization is more likely to enter the electrode, which leads to the insignificance of the charge peak and the lower accumulation of space charge.

Compared with pure polyethylene, the heteropolar space charge near the electrode was increased after the doping microcapsule. There were many small charge peaks, which gradually increased and shifted to the inside of the sample. Furthermore, the mean volume density of space charge in the composite was still at a lower level.

A doping microcapsule system can increase the number of polar groups, enhancing the ionization, and thus increasing the accumulation of heteropolar space charge. On the one hand, new charge traps can be introduced by the interface area between the microcapsule and matrix (as shown in Figure 8). On the other hand, the microcapsule increases the number of grains in the polyethylene (Section 3.2.1), so the interface between the crystal and amorphous regions is increased, which increases the number of deep charge traps. Therefore, doping microcapsules can increase the number and depth of charge traps in polyethylene, which enhances the ability of the material to bind to space charge. Because most of the new crystal regions are around the microcapsule, the distribution of charge traps in the composite is related to that of microcapsule (as shown in Figure 1), resulting in more charge peaks. In addition, the composite has a higher binding effect on charge and a higher volume resistance (Section 3.2.2), which not only inhibits the carrier injection, but also hinders the movement of space charge, enhancing the neutralization of ionization products. Thus, the amount of space charge in the composite is low.

In conclusion, although doping microcapsules can introduce new impurities to enhance ionization, the number and depth of space charge traps in the polyethylene can be increased by the interface between the microcapsule and matrix. Doping microcapsules can not only inhibit the injection of carriers at the electrode, but also restrict the movement of charge, enhancing its neutralization, thus reducing the space charge in the composite. Therefore, the space charge characteristics of the composite can meet the basic application requirements of polyethylene insulating material.

## 4. Conclusions

In this paper, the preparation method and typical electrical performances of the novel self-healing polyethylene insulating composite were explored exploitatively and creatively. On the basis of proving the self-healing ability of the composite, the effect mechanisms of the doping microcapsules on the important electrical performances of polyethylene were studied emphatically. The application advantages of self-healing insulating composites were proven by linking the micro-mechanism with the macro-performance. The main conclusions are as follows:

(1) The microcapsule/polyethylene insulating composite prepared in this paper has obvious self-healing abilities against structural damage. Moreover, according to the repair results of breakdown strength, the appropriate repair conditions selected in this paper were 60 °C/30 min. In these conditions, the performances of the composite hardly changed, but the self-healing effect was remarkable. The composite can not only fill the structural defect caused by mechanical damage (such as a scratch) and electrical damage (such as an electrical tree), but also homogenize the local high electric field in the defect area, restoring its insulation strength;

(2) Doping microcapsules can directly change the basic physicochemical properties (such as crystallinity) of polyethylene, and then affect the electrical performance. Microcapsules increase the crystallization source of polyethylene by the heterogeneous nucleation effect of the interface between the microcapsule and matrix. However, the microcapsule cannot change the size of the original crystal region in matrix, and the modified process for doping microcapsules (reducing pressure and prolonging heating time) can decrease the number and size of crystal grains in the polyethylene;

(3) The effect of doping microcapsules on the electrical performance of polyethylene, such as breakdown strength, volumetric resistivity, dielectric properties, and space charge characteristics, are mainly related to impurity (i.e., contaminant) and the interface. The polarization and ionization of the impurities introduced by the microcapsule system can reduce the electrical performance of the polyethylene. However, the interface between the microcapsule and matrix can not only improve the microstructure of the matrix (such as the heterogeneous nucleation effect increasing the number of crystal regions, and the anchoring effect enhancing the stability of the amorphous regions), but also increase the space charge trap. Moreover, the microstructure and charge trap can affect the characteristics of carrier transport, material polarization, and space charge accumulation, thus improving the electrical performance of polyethylene to a certain extent.

In addition, the typical electrical performance of the self-healing polyethylene composite in this paper can meet the basic application requirements of insulating material. It can also be predicted that the composite has high application value in the appropriate concentration range of the microcapsules.

**Author Contributions:** Conceptualization, Y.W. and Y.L.; Methodology, Y.L.; Validation, Y.W., Y.L. and Z.Z.; Data curation, Y.Z. and Z.Z.; Formal analysis, Y.W. and Y.L.; Writing—original draft preparation, Y.L.; Writing—review and editing, Y.W.; Visualization, Y.L. and Y.W.; Project administration, Y.W.

**Funding:** This research was funded by National Natural Science Foundation of China (51777018).

**Conflicts of Interest:** The authors declare that they have no conflict of interest.

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
