# Peer review of "Effect of Doping Microcapsules on Typical Electrical Performances of Self-Healing Polyethylene Insulating Composite"

_applsci, doi:10.3390/app9153039_

Round 1
Reviewer 1 Report
I think the manuscript is well-written and organized, and provides significant information on the preparation of polyethylene composites and their properties. Please check the attached file to find my comments.

Reviewer 2 Report
This manuscript showed systematically investigate an insulation property of the novel self-healing polyethylene insulating composite. Especially, it is very interesting to improve insulation property by self-healing effect. However, I think that this study have to need additional experimentals be accepted as follows. 1. The author showed that composite can not only repair the electrical tree and scratch, but also restore the insulation strength of damaged area. It was explained that doping microcapsule on the polyethylene electrical performances are mainly related to impurity and interface from microcapsule. I think that additional test depend on microcapsule ratio in composite is required to demonstrate. 2. The insulating property of composite can be changed by heating temperature. However, author only treat at 60℃ for 30 min because of melting temperature of repairing agent. I think that author should give insulating property of self-healing polyethylene insulating composite depend on treatment condition (ex) temperature and time).
Reviewer 3 Report
Electrical performances of the self healing polyethylene insulating composite are important.
Additionally, emphatical studies are cost effective. These performances of self-healing polyethylene composite are useful in meeting the application of insulating materials.
The authors are to be commanded for their excellent work.
Round 2
Reviewer 2 Report
Authors fully explained about my review in anwer sheet